# DORT: Modeling Dynamic Objects in Recurrent for Multi-Camera 3D Object Detection and Tracking

**Qing Lian**[1,2]  **Tai Wang**[1,3]  **Dahua Lin**[1,3]  **Jiangmiao Pang**[1✉]

[1]Shanghai AI Laboratory  [2]The Hong Kong University of Science and Technology
[3]The Chinese University of Hong Kong

qlianab@connect.ust.hk, {wt019,dhlin}@ie.cuhk.edu.hk, pangjiangmiao@gmail.com

**Abstract:** Recent multi-camera 3D object detectors usually leverage temporal information to construct multi-view stereo that alleviates the ill-posed depth estimation. However, they typically assume all the objects are static and directly aggregate features across frames. This work begins with a theoretical and empirical analysis to reveal that ignoring the motion of moving objects can result in serious localization bias. Therefore, we propose to model Dynamic Objects in RecurrenT (DORT) to tackle this problem. In contrast to previous global Bird-Eye-View (BEV) methods, DORT extracts object-wise local volumes for motion estimation that also alleviates the heavy computational burden. By iteratively refining the estimated object motion and location, the preceding features can be precisely aggregated to the current frame to mitigate the aforementioned adverse effects. The simple framework has two significant appealing properties. It is flexible and practical that can be plugged into most camera-based 3D object detectors. As there are predictions of object motion in the loop, it can easily track objects across frames according to their nearest center distances. Without bells and whistles, DORT outperforms all the previous methods on the nuScenes detection and tracking benchmarks with 62.8% NDS and 57.6% AMOTA, respectively. Codes are available at https://github.com/OpenRobotLab/DORT.

**Keywords:** Temporal Modeling, 3D Object Detection

## 1 Introduction

Multi-camera 3D object detection is critical to robotic systems such as autonomous vehicles, humanoid robots, and *etc.* As object depth estimation from a single image is naturally ill-posed, recent works use large-scale depth pre-trained models [1] and leverage geometric relationships [2, 3, 4, 5] to alleviate the problem. Because stereo correspondence exists in consecutive frames, some works resort to temporal information for accurate depth predictions. For example, BEVDet4D [6] and BEVFormer [7] warp preceding features to the current frame to enrich the single-frame BEV representations. DfM [5] constructs temporal cost volumes that explicitly establish the stereo correspondence. However, these cross-frame feature aggregations do not consider the motion of moving objects and assume all the objects are static, which results in serious 3D localization bias.

In this paper, we first provide a theoretical and empirical analysis to reveal the negative effects of inaccurate object motion to object depth (Fig 1). In particular, if the object is moving, the incorrect temporal correspondence would derive a biased depth. In the driving scenarios, it is critical that a misleading depth is estimated, which might reduce the reaction time of the decision system, leading to catastrophic collision accidents. This motivates us to devise an explicit mechanism to involve object motion estimation in the temporal-based 3D detection pipeline.

---

✉Corresponding author.

7th Conference on Robot Learning (CoRL 2023), Atlanta, USA.

Modeling dynamic objects in this context has several challenges: (1) We need a flexible object-wise representation for potential object-wise operations based on motion modeling. (2) Jointly estimating object location and motion is an inherent chicken and egg problem [8]: The temporal correspondence can derive accurate object location only when accurate object motion is given and vice versa. (3) Simultaneously predicting object location and motion from only two frames is also an ill-posed problem theoretically, and thus it is desired to involve right-body assumption and more frames to pose reasonable constraints.

To address these problems, we model **D**ynamic **O**bjects in **R**ecurren**T** (DORT) that simultaneously estimates object motion and location, and then progressively refines them for accurate 3D object detection. It benefits from a local 3D volume representation that not only extracts object-wise 3D features but also alleviates the heavy computational costs of global BEV in previous methods [5, 7, 9]. Based on the object-wise volume, temporal volumes are constructed by warping the volumes from the preceding frame to the current frame according to the object motion. Then the obtained cost volumes act as the features for updating the candidate location and motion. We model this estimation and update pipeline as a recurrent process to alleviate the aforementioned chicken and egg problem. In addition, our framework can take into more than two frames and pose constraints to the object motion. It inherently provides a feasible solution to avoid the ill-posed dilemma of estimating object location and motion from only a single pair of correspondence.

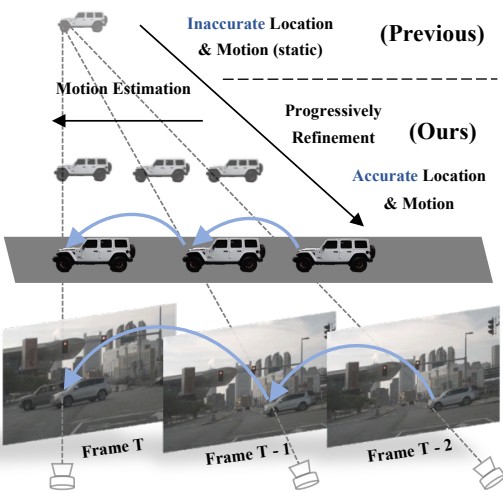

Figure 1: Visualization of object localization from temporal correspondence. Previous works ignore the motion of moving objects, leading to imprecise localization. Our work progressively refines the object's location and motion so that the preceding features can be precisely aggregated.

As there is object motion prediction in the loop, the framework is naturally capable of joint object detection and tracking by utilizing object motion to align the detection results into the same timestamp. It also can be plugged into most camera-based 3D object detectors for flexible and practical use.

We validate the effectiveness of our framework on the nuScenes detection and tracking benchmarks. Benefiting from the dynamic objects modeling, DORT outperforms all the previous methods with a large margin, leading to 62.8% nuScenes detection metric (NDS) and 57.6% and average multi-object tracking accuracy (AMOTA), respectively.

## 2    Related work

**Monocular 3D Object Detection**    Monocular-based 3D object detection was first approached from the single-frame scenario and evolved into multi-frame to alleviate the ill-posed depth estimation.

*(a) Methods with A Single Frame*    The single-frame-based methods [10, 11, 3, 12, 13] first extend 2D object detectors and insert several 3D attribute regression heads to predict 3D bounding boxes. To alleviate the ill-posed depth recovery, several methods improve the model from the perspectives of loss function [11], network architecture [14, 15], regression objective [3, 4], *etc.* Besides directly regressing depth, later approaches [2, 16, 17] further design 2D-3D geometry constraints to better extract visual cues for depth estimation. To align detection features with the output space, another line of methods [18] designs several transformation modules to lift 2D inputs into 3D space. Pseudo-lidar-based methods [19, 20, 21, 21] first predict the per-pixel depth and convert the raw pixel into point cloud for 3D detection. BEV-based methods [18, 22, 9, 23] propose orthographic feature transformation (OFT) to transform the 2D features into 3D voxels and then adopt a LiDAR-based

head to localize objects. Later works improve the OFT by explicit depth distribution modeling [22, 9, 23], incorporating deformable attention module [7] or designing 3D position encoding [24, 25].

*(b) Methods with Multiple Frames* Although many techniques are designed in single-frame-based methods, they still suffer from ill-posed depth recovery, leading to unsatisfactory performance for deployment. To augment the single-view observation, recent works [26, 14, 5, 6, 7, 27] leverage previous frames as additional observations for features augmentation. Kinematic3D [14] leverages 3D Kalman Filter to associate objects across frames and refines 3D boxes. Later studies [5, 6, 7] construct cross-frame cost volumes as another visual cue for 3D detection. The cost volumes are based on the multi-view stereo, which assumes objects are static across frames. However, this assumption does not align with the driving scenario, where the objects can move.

**Monocular 3D Object Tracking** 3D object tracking associates objects across frames and generates a set of trajectories for motion prediction. Traditional methods adopt a tracking-by-detection paradigm that first detects objects in each frame and then associates them by appearance features [26] or objects' displacement with Kalman filter [28, 29, 30, 31, 32, 33]. Besides the above paradigm, several methods [34, 35] design a two-stage paradigm that first associates objects and then utilizes the temporal motion to improve the detection performance. In this work, we utilize temporal cost volumes to bridge the spatial location and temporal motion and derive a recurrent paradigm that iteratively updates them to obtain tightly coupled results for joint 3D detection and tracking.

## 3  Object Motion in Temporal Modeling

In this section, we first conduct analysis to demystify the adverse effects of neglecting object motion in temporal modeling, then discuss the challenges of modeling 3D motion in the monocular setting.

**Localization Bias from the Static Assumption** In previous methods [5, 6, 7], the object motion is ignored by assuming objects are static and the features are directly aggregated after converting the past frames to the current frame. We first show that the static assumption would derive a biased depth. Without loss of generality, we consider the two-view case, and it can be naturally extended to more than two views. We denote the camera intrinsic as $K$ with focal length and center offset $(f, c_u, c_v)$, and the ego motion and object motion from frame $t_0$ to frame $t_1$ as $T_{t_0 \to t_1}^{ego}$ and $T_{i \to j}^{obj}$:

$$K = \begin{bmatrix} f & 0 & c_u \\ 0 & f & c_v \\ 0 & 0 & 1 \end{bmatrix}, T_{t_0 \to t_1}^{ego} = \begin{bmatrix} 1 & 0 & 0 & x^{ego} \\ 0 & 1 & 0 & 0 \\ 0 & 0 & 1 & z^{ego} \end{bmatrix}, T_{i \to j}^{obj} = \begin{bmatrix} 1 & 0 & 0 & x^{obj} \\ 0 & 1 & 0 & 0 \\ 0 & 0 & 1 & z^{obj} \end{bmatrix}. \quad (1)$$

For simplicity, we assume the ego and object motion only contain the translation $(x, 0, z)$ on the horizontal plane. The analysis can be easily extended to the case that the motion contains rotation. Given the temporal images, we can utilize photometric or featuremetric similarity to find the correspondence of pixel $p_{t_0} = (u_{t_0}, v_{t_0})$ in the frame $t_0$ and pixel $p_{t_1} = (u_{t_1}, v_{t_1})$ in the frame $t_1$. Then, the depth $z_{t_1}$ can be recovered by:

(a) Average depth error vs object velocity    (b) Histogram of object velocities

Figure 2: Empirical analysis of the depth bias on the nuScenes dataset if objects are assumed static.

$$T_{t_0 \to t_1}^{ego} \cdot \pi(p_{t_0}, K) = \pi(p_{t_1}, K), \quad z_{t_1} = \frac{z^{ego}(u_{t_0} - c_u) - f x^{ego}}{u_{t_0} - u_{t_1}}, \quad (2)$$

where $\pi$ denotes the 2D to 3D projection. This derivation assumes that the object is static. However, objects can move with a corresponding motion $T_{t_0 \to t_1}^{obj}$. Then, the object depth is revised as follows:

$$T_{t_0 \to t_1}^{obj} T_{t_0 \to t_1}^{ego} \cdot \pi(p_{t_0}, K) = \pi(p_{t_1}, K), \quad \hat{z}_{t_1} = \frac{(z^{ego} + z^{obj})(u_{t_0} - c_u) - f(x^{ego} + x^{obj})}{u_{t_0} - u_{t_1}}. \quad (3)$$

Based on Eq (2) and (3), we can obtain the depth gap:

$$\Delta z = \frac{z^{obj}(u_{t_0} - c_u) - f x^{obj}}{u_{t_0} - u_{t_1}}. \quad (4)$$

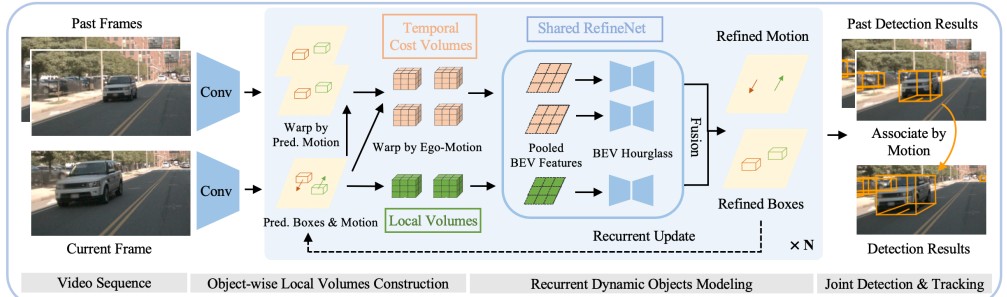

Figure 3: Pipeline overview. Given a video sequence, we first extract the 2D features and generate the candidate boxes and motion by a single-frame detector. Then the boxes and motion are progressively refined from the concurrently updated 3D volume features. A fusion process in the recurrent module combines the estimation from each pair of frames. Based on the tightly coupled modeling of object location and motion, the framework can achieve joint 3D detection and tracking during inference.

From Eq (4), we can observe that the depth bias is linearly correlated with the object motion. In Fig 2, we also display the empirical statistics of object motion and the corresponding depth bias from the nuScenes dataset. We can observe that the empirical depth error is also correlated with the object velocity and increases as the time interval enlarges. Besides, the right part in Fig. 2 also shows that almost 51% of objects are moving across frames, demonstrating the necessity of modeling object motion in the temporal-based framework.

**Ill-Posed Problem in Motion Modeling** Except for demonstrating the necessity of modeling object motion, we also want to mention that simultaneously estimating object location and motion is nontrivial, especially in the two-frame case. As shown in Fig 1, the correspondence of two points can come from infinite combinations of object location and motion. This illustrates that joint location and motion estimation from only one correspondence is ill-posed. To alleviate this issue, we first simplify the object motion as a right-body movement so that multiple correspondences from the points in the object can be used to solve a shared motion. Furthermore, we leverage more than two frames to constrain the flexibility of object motion. More details can be referred to Sec 4.3.

# 4 Methodology

This section describes the details of DORT. DORT is a general joint detection and motion prediction module that can estimate coupled object location and motion results across frames. Based on the tightly coupled location and motion results, DORT is also capable of simultaneously 3D object detection and tracking. Basically, it can be based on most temporal 3D detectors [6, 36, 7]. In this work, we select the popular temporal detector BEVDepth [23] as the base detector and extend it to handle both static and moving objects in temporal modeling. We first present an overview of temporal-based frameworks in Sec 4.1 and then introduce our modifications: the local volume for object-wise representation in Sec 4.2, the key recurrent dynamic objects modeling in Sec. 4.3, and the object association for monocular 4D object detection in Sec 4.4.

## 4.1 Overview of Temporal-Based Frameworks

Current temporal-based methods contain three stages: (1) The 2D features extraction stage extracts the features from the 2D images. (2) The view transformation and stereo matching process that first lift the 2D features to a 3D volume and then warp the features in each frame to an aligned canonical space for matching. Depending on the model design, the order of view transformation and stereo matching may reverse. (3) The detection stage takes the 3D features to estimate 3D bounding boxes.

In this work, we follow previous methods [5, 9, 23] and adopt the widely-used 2D backbone (*e.g.* ResNet [37]) to do the features extraction. For the view transformation stage, we design an **object-wise local volume** that leverages the candidate 3D boxes to obtain the potential foreground regions and only models them with local object-wise 3D volumes. For the stereo matching and detection

stages, we propose a **recurrent dynamic objects modeling** module to progressively refine the detection and motion results for accurate 3D temporal features.

## 4.2 Object-wise Local Volume

In previous works, the 2D-3D transformation considers each candidate 3D grid point and constructs a global volume for detection. However, there are several limitations: (1) the global volume contains lots of background regions, which is not vital for detection but increases the computation burden. (2) Modeling a global volume needs to pre-define a detection range during training, making the detectors fail to detect objects with arbitrary depth. (3) It is inconvenient to manipulate a global volume with object-wise operations.

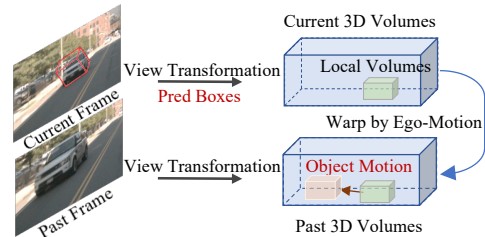

Figure 4: The process of extracting local volumes in the current and past frames according to predicted bounding boxes and object motion.

Hence, we replace the global volume with an object-wise local volume. Specifically, we leverage the candidate boxes to determine the 3D region of interest (RoI) and set the local volume center as the bounding box center. To keep the object ratio and achieve cross-view warping, we assign each 3D RoI volume $V \in \mathcal{R}^{W \times H \times L \times C}$ with the same 3D dimension $(W, H, L)$ and channel size $C$. Different from 2D detection, the objects' dimension in 3D space has less variance and empirically relies less on the RoI-Align [38] operation. We display the construction of object-wise local volumes in Fig 4. For the 2D to 3D transformation, we first follow LiftSplat [22] and lift the images to a 2.5D frustum by weighting with depth probability. Then we utilize the grid sample operation to warp the features from the 2.5D frustum to each 3D local volume. Benefiting from the accurate 2D detection performance, the local volume features sampled from the 2.5D frustum would have a large overlap with the foreground objects. Hence, if the proposal 3D location is inaccurate, the later refinement module still can use the features for refinement.

## 4.3 Recurrent Dynamic Objects Modeling

The pipeline of the recurrent framework is illustrated in Fig. 3. Given the candidate 3D bounding boxes and motion as input, each iteration first constructs the temporal cost volumes thereon, and aggregates these cues to refine the proposal boxes and motion. In particular, we adopt a perspective-view based 3D detector (*i.e.* PGD) to generate the initialized candidate 3D boxes and motion and only predict their residuals for refinement in the subsequent recurrent updates.

**Cross-Frame Cost Volumes Construction** Given the initial predictions of 3D boxes and their motion, we first obtain object-wise volume features following Sec. 4.2. Then we can construct the temporal cost volumes by warping features from past frames to the current frame coordinates based on *ego-motion*. In contrast to previous works [5, 6] assuming objects are static, we further involve the *object motion* into the warping procedure. Specifically, for each point $p \in \mathcal{R}^3$ in the object-wise local volume $V$, we query the corresponding features in previous frame $t - \Delta t$ with the consideration of ego-motion $T^{ego}$ and the object motion $T^{obj}$ and construct the cost as $\left[V(p), V_{t-\Delta t}(T^{obj}T^{ego}p)\right]$. Note that we simplify the point motion as the object motion with a rigid-body assumption, which can approximate most of the cases in driving scenarios, especially for vehicles [26, 39].

**3D Boxes and Motion Residual Estimation** Given the object-wise temporal features built from input 3D boxes and motion, we leverage a refinement network to estimate the residual between the input 3D boxes and motion with the ground truth. The refinement network contains several 2D/3D residual-based convolutional layers to extract the 3D volumes and 2D BEV features. The detailed architecture is presented in Supplementary. Formally, the refinement is formulated as the regression of 3D attribute residuals $\mathcal{B}$, including the object's 3D center $x, y, z$, 3D size $w, h, l$, rotation $\theta$ and velocity $v_x, v_y$. Since we use the object velocity in the current frame to represent the object motion and assume constant velocity across frame, the supervision for some frames may contain noise(*e.g.* inaccurate labels, violation of rigid-body assumption, etc). Hence, we model the residual

Table 1: Experimental results of monocular 3D object detection and tracking on the nuScenes test set. The input resolution is $1600 \times 900$ with using ConvNeXt-Base [40] as the backbone.

(a) 3D detection results.

| Method | mAP↑ | mATE↓ | mASE↓ | mAOE↓ | mAVE↓ | mAAE↓ | NDS↑ |
|---|---|---|---|---|---|---|---|
| Ego3RT [41] | 42.5 | 0.55 | 0.26 | 0.43 | 1.01 | 0.14 | 47.3 |
| UVTR [42] | 47.2 | 0.57 | 0.25 | 0.39 | 0.51 | 0.12 | 55.1 |
| BEVFormer [7] | 48.1 | 0.58 | 0.25 | 0.37 | 0.37 | 0.12 | 56.9 |
| PETRv2 [43] | 51.2 | 0.55 | 0.25 | 0.36 | 0.40 | 0.13 | 58.6 |
| BEVDepth [23] | 52.0 | 0.45 | 0.24 | 0.35 | 0.35 | 0.13 | 60.9 |
| BEVStereo [27] | 52.5 | 0.43 | 0.24 | 0.36 | 0.35 | 0.14 | 61.0 |
| SOLOFusion [36] | 54.0 | 0.45 | 0.26 | 0.37 | 0.27 | 0.14 | 61.9 |
| DORT (Ours) | **55.3** | 0.43 | 0.26 | 0.42 | **0.24** | 0.14 | **62.8** |

(b) 3D tracking results.

| Method | AMOTA↑ | AMOTP↓ | MOTAR ↑ |
|---|---|---|---|
| QD-3DT [34] | 21.7 | 1.550 | 56.3 |
| Time3D [35] | 21.4 | 1.360 | - |
| PolarDETR [30] | 27.3 | 1.185 | 60.7 |
| MUTR3D [44] | 27.0 | 1.494 | 64.3 |
| SRCN3D [31] | 39.8 | 1.317 | 70.2 |
| QTTrack [32] | 48.0 | 1.100 | 74.7 |
| UVTR [42] | 51.9 | 1.125 | 76.4 |
| DORT(Ours) | **57.6** | **0.951** | **77.1** |

as a Laplacian distribution and design the loss function as:

$$\mathcal{L}_{refine} = \sum_{b \in \mathcal{B}} (\frac{\sqrt{2}}{\sigma_b} \|\Delta\hat{b} - \Delta b\| + \log \sigma_b). \tag{5}$$

Here, $\Delta b$, $\Delta\hat{b}$, and $\sigma_b$ are all the network outputs, and represent the ground truth residual, the estimated residual, and the estimated standard deviation of residual for each 3D attribute, respectively.

**Multiple Estimation Fusion** Given $n$ frame as inputs, we can obtain $n$ 3D volumes (1 for the local volumes from the current view and $n - 1$ for the paired cross-view cost volumes) and obtain $n$ estimated residuals from the residual estimation module. Then we weigh the importance of each residual by the estimated deviation and fuse them to obtain an ensemble result:

$$\hat{b}_{fused} = \sum_{i=1}^{n} \frac{e^{\sigma_{b_i}} b_i}{\sum_{i=1}^{n} e^{\sigma_{b_i}}}, \tag{6}$$

where $i$ denotes the volume index. For simplicity, here we only estimate the velocity measurement for the referenced frame, *i.e.*, the fluctuation of object velocity across different frames would not be considered explicitly. The mechanism for multi-frame fusion is expected to handle this problem adaptively. At the same time, this constraint also provides additional cues when simultaneously estimating object location and motion from more than two frames.

**Recurrent Location and Motion Update** After each iteration, we can obtain the refined bounding boxes with their motion and thus can derive the updated bounding boxes in different frames. With these updated locations and RoIs, we can further update the volume features and proceed to the next-round refinement. Note that any complex or learnable motion modeling can be integrated into this procedure. Here, to be consistent with the multiple estimation fusion designs, we still keep the constant velocity prediction for simplicity to derive the bounding boxes of previous frames.

In the training stage, we follow the recurrent methods [45, 39] in other tasks and set the loss weight for each iteration as the same. The overall loss is represented as:

$$\mathcal{L} = \mathcal{L}_{pv} + \sum_{i=1}^{k} \mathcal{L}_{refine}^i, \tag{7}$$

where $\mathcal{L}_{pv}$ is the loss in the perspective-view detector [4], $\mathcal{L}_{refine}^i$ is the refinement loss in each iteration and $k = 3$ is the number of the iterations. In the inference stage, we first take the perspective-view detector (*i.e.* PGD [4]) to generate the initial 3D boxes and their motion and then progressively refine them. In each iteration, we first construct the volume features as discussed in Sec 4.3 and feed them into the refinement module to estimate the 3D boxes and motion residuals for each paired frame input. Then we utilize the multiple estimation fusion module in Sec 4.3 to fuse the estimated results and obtain the refined 3D boxes and motion as the next stage input.

### 4.4 Monocular 4D Object Detection

So far, we have introduced our recurrent framework for 3D detection from monocular videos. Based on progressive refinement, our model can estimate tightly coupled object location and motion results and thus can easily associate the object detection results across frames, leading to joint 3D detection

Table 2: Experimental results on the nuScenes validation set. The input resolution is $704 \times 256$ using ResNet-50 as the backbone. * denotes the re-implementation based on the provided code.

| Methods | # frame | mAP↑ | mATE↓ | mASE↓ | mAOE↓ | mAVE↓ | mAAE↓ | NDS↑ |
|---|---|---|---|---|---|---|---|---|
| PGD* [4] | | 28.8 | 0.75 | 0.27 | 0.52 | 1.13 | 0.18 | 37.0 |
| BEVDet [9] | 1 | 29.8 | 0.73 | 0.28 | 0.59 | 0.86 | 0.24 | 37.9 |
| PETR [25] | | 31.3 | 0.77 | 0.28 | 0.56 | 0.92 | 0.23 | 38.1 |
| DETR3D [24] | | 34.9 | 0.72 | 0.27 | 0.38 | 0.84 | 0.20 | 43.4 |
| BEVDet4D [6] | | 32.2 | 0.70 | 0.28 | 0.50 | 0.35 | 0.21 | 45.7 |
| BEVDepth [23] | 2 | 35.1 | 0.64 | 0.27 | 0.48 | 0.43 | 0.20 | 47.5 |
| DORT (Ours) | | **37.9** | 0.62 | 0.27 | 0.45 | **0.31** | 0.20 | **50.4** |
| BEVDepth* [6] | 8 | 39.8 | 0.57 | 0.27 | 0.49 | 0.27 | 0.18 | 52.3 |
| DORT (Ours) | | **41.8** | 0.57 | 0.26 | 0.43 | **0.25** | 0.19 | **53.4** |
| SOLOFusion [36] | 16 | 42.7 | 0.57 | 0.27 | 0.41 | 0.25 | 0.18 | 53.4 |
| DORT (Ours) | | **43.6** | 0.56 | 0.26 | 0.41 | **0.24** | 0.18 | **54.0** |

and tracking. Specifically, we follow [28, 46, 29, 47] and associate the detection results by warping current detection to the past frames with object motion. Based on the ego-motion, we first convert the predicted object location to the past frame coordinate and then warp with the estimated object velocity. Then we follow the popular distance-based tracker [28, 29] and associate the objects by the closest distance matching. We provide more details of the tracking pipeline in the Appendix.

## 5 Experiments

### 5.1 Experimental Setup

In this section, we describe the used dataset, the evaluation metrics, and the implementation details.

**Dataset** NuScenes [48] is a large-scale driving dataset, which contains 1,000 video sequences. The official protocol splits the video sequences into 700 for training, 150 for validation, and 150 for testing. Each sequence is annotated with the objects' 3D bounding box, velocity, and tracking id.

**Network Details** As discussed in Sec 4.3, the recurrent module requires a proposal detector to generate candidate foreground regions as the $1^{st}$ stage input. We adopt the popular monocular 3D detector PGD [4] due to its high 2D object detection recall. Following [9, 6, 23], we adopt the ResNet-50 [37] with FPN as the 2D feature extractor and mainly conduct experiments on this setting. The 2D feature extractors in PGD and the recurrent module are shared to save computation time. The grid size of the 3D volume is set as $0.8m$ with the range of $[-5m, 5m]$ in the X and Z (depth) axis and $[-4m, 2m]$ in the Y (height) axis. During 2D to 3D features transformation, we follow [23] and adopt the depth distribution guided 2D to 3D features lifting. Regarding the test set submission, we follow [23, 27] and adopt the ConvNeXt-Base [40] as the image backbone. The image backbone is initialized with ImageNet pre-trained weights, and no other external data is used. We provide more details about the network architecture of the recurrent module and the training configuration in the supplementary materials.

**Training Configurations** The model is optimized by AdamW optimizer with weight decay $10^{-2}$. We first follow [4] to train the proposal detector and refine the recurrent module with 24 epochs, where the initial learning rate is set as $2 \times 10^{-4}$ and decreases to $2 \times 10^{-5}$ and $2 \times 10^{-6}$ at the $18^{th}$ and $22^{th}$ epochs. Following [9, 23], we use the class balance sampling strategy (CBGS) to alleviate the class imbalance problem. We adopt the commonly used 2D data augmentation that randomly flips the image, resizes the image with the range of $[0.36, 0.55]$, and crops the image to the resolution of $704 \times 256$. Regarding the input video sequences, we follow [6, 23] and sample the preceding keyframes to obtain the past video sequences. Regarding the test set submission, we enlarge the input resolution to $1600 \times 640$ and reduce the volume size to $0.4m$.

### 5.2 Main Results

In Table 1, we first provide the comparison of our framework with existing state-of-the-art methods on the nuScenes test benchmarks. We draw the following observations: (i) Benefiting from dynamic objects modeling, our method displays a significant improvement in both object detection (mAP) and motion estimation (mAVE), and 1.3% and relatively 11.1% better than the previous best method. These localization and motion estimation improvements also contribute to state-of-the-art results in

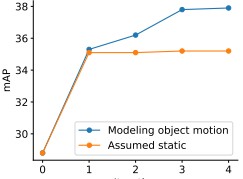 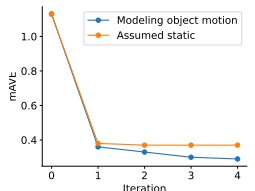

| Setting | mAP | NDS | mAVE |
|---|---|---|---|
| Assumed static | 35.0 | 47.1 | 0.37 |
| GT motion | 39.3 | - | - |
| Pred motion | 37.9 | 52.1 | 0.31 |

Figure 5: Left and Middle: mAP and mAVE in each recurrent iteration step (1 past frame is used). Right: Experimental results of different motion modeling strategies.

terms of the nuScenes detection metric (NDS). (ii) With strong localization and motion estimation results, our tracking module can better associate the detected objects in different timestamps, resulting in superior performance over all the other trackers with different metrics. Specifically, we improve the second best tracker [42], another distance-based tracker with % and 7.5% relative improvements on the AMOTA and AMOTP metrics. Compared with the joint detection and tracking methods QDTrack3D [34] and Time3D [35], the performance gain of our method demonstrates the effectiveness of our dynamic objects modeling framework in jointly modeling object motion and location. (iii) In Table 2, we also report our method on the nuScenes validation set with different settings. For the detection performance, we can draw the same observation as in the test set that our method can outperform previous temporal-based methods [6, 23, 36] in terms of mAP and mAVE. Note that our method is also compatible with the components designed in the current BEV-based frameworks, such as the training techniques in BEVDepth [23] and the depth estimation module in SOLOFusion [36]. Furthermore, the local 3D volume is more friendly to practical applications, which can handle objects with arbitrary depth in the image.

## 5.3 Ablation Study

**Ablation Study of Object Motion** We further validate the influence of different dynamic object modeling strategies on the detection performance The first experiment compares the assumed static case with that of using the ground truth object motion. As shown in Figure 5, the model with ground truth object motion outperforms the assumed static with 4.3% mAP, demonstrating the necessity of object motion for obtaining accurate temporal correspondence features. When we replace the ground truth object motion with an estimated one, it still can bring 2.9% mAP improvements, illustrating the usefulness of our dynamic objects modeling module.

**Experiments with Different Iterations** In Figure 5, we provide the comparison of modeling object motion and assumed static with different recurrent iterations. Benefiting from the BEV features modeling, the two configurations display almost 2% mAP improvements in the first iterations. In the later iterations, the improvement in assumed static stops was mainly due to the lack of accurate temporal features. With more and more accurate temporal features, the model with modeling object motion can progressively improve the detection and motion estimation results.

## 6 Conclusion and Limitations

This work proposes a novel framework to better leverage temporal information for camera-only 3D detection by modeling dynamic objects. We first design an object-wise local volume to save computation time and maintain an object-wise representation for motion and detection modeling. Then we propose a recurrent module to tackle the challenging motion and location modeling problem. Specifically, we progressively update the motion and location results from the concurrently updated 3D volume features. As the object motion and location results are tightly coupled in the recurrent stage, we also demonstrate the framework can naturally achieve 3D tracking.

Although DORT can better handle dynamic objects, we follow previous methods and simplify the motion with a constant velocity assumption. To handle various kinds of scenarios, this assumption could be relaxed with modeling acceleration or explicit trajectory prediction. Additionally, similar to current object-wise methods [24, 25, 49] (*i.e.* DETR-based, Two-stage-based), the computation is linearly correlated with the number of instances. To overcome this limitation, point-wise motion modeling and feature extraction with object-wise grouping will be considered in future work.

**Acknowledgement** This project is supported by Shanghai Artificial Intelligence Laboratory.

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

# Supplementary Material for DORT: Modeling Dynamic Objects in Recurrent for Multi-Camera 3D Object Detection and Tracking

## A  Evaluation Metrics

**Detection Metrics**  We adopt the official evaluation protocol provided by nuScenes benchmark [48]. The official protocol evaluates 3D detection performance by the metrics of average translation error (ATE), average scale error (ASE), average orientation error (AOE), average velocity error (AVE), and average attribute error (AAE). Besides, it also measures the mean average precision (mAP) with considering different recall thresholds. Instead of using 3D Intersection over Union (IoU) as the criterion, nuScenes defines the match by 2D center distance $d$ on the ground plane with thresholds $\{0.5, 1, 2, 4\}m$. The above metrics are finally combined into a nuScenes Detection Score (NDS).

**Tracking Metrics**  Regarding the tracking metrics, the nuScenes benchmark mainly measures the average multi-object tracking accuracy (AMOTA), average multi-object tracking precision (AMOTP), and tracking recall. In particular, AMOTA and AMOTP are the averages of multi-object tracking accuracy (MOTA) and multi-object tracking precision (MOTP) under different recall thresholds.

## B  Implementation Details

In the main paper, we have introduced our overall multi-camera 3D object detection and tracking framework and the details of the proposed components. In this supplemental section, we present the details of the other basic modules.

### B.1  Network Architecture

Our framework is built based on BEVDet and BEVDepth, and we follow them to design the basic modules.

**2D Feature Extraction**  Given $N$ multi-view images $I \in \mathcal{R}^{N \times W \times H \times 3}$ in each frame, we use a shared 2D backbone to extract the corresponding features. We adopt the standard ResNet-50 [37] as the backbone and initialize it with ImageNet pre-trained weights. Then we adopt a modified Feature Pyramid Network (FPN) [50] to extract the multiple-level features and the output 2D features are downsampled with the ratio of $\frac{1}{16}$ with channel size 256: $F_{pv} \in \mathcal{R}^{\frac{W}{16} \times \frac{H}{16} \times 256}$.

**View Transformation**  Our work is the same as BEVDet and BEVDepth which contains a 2D to 3D view transformation module. Specifically, we first leverage a depth prediction head to predict the depth probability for each pixel. Then we lift the 2D features to a 2.5D frustum space via out-product it with the depth probability. The depth probability range is set as $[0m, 60m]$ with grid size $0.5m$. With the 2.5D frustum features, the 3D features for each local volume are obtained via utilizing the camera intrinsic to project the 3D grid back to the frustum and bi-linear sample the corresponding features. As mentioned in the main paper, we aggregate the 3D volume features along the height dimension and obtain the corresponding object-wise BEV features $F_{bev}^{obj} \in \mathcal{R}^{N \times W^{obj} \times H^{obj} \times 256}$, where $W^{obj}$ and $H^{obj}$ are the object features dimension and set as 28 in the main setting.

**RefineNet**  Given the object-wise features extracted based on the proposal 3D box and motion, RefineNet takes several convolutional neural networks to extract the object-wise features and estimate the bounding box and motion residual. Specifically, we first adopt an average pooling layer to aggregate the 3D features along the height dimension and obtain the BEV features. Then we filter each object-wise BEV features with 6 basic 2D residual blocks, where each residual block consists

of two 2D convolution layers and a skip connection module as in ResNet. The channel size of the residual blocks in the first three layers is 256 and decreases to 64 in the last three layers. Then we aggregate the features along the spatial dimension via average pooling and take 4 layers MLP network to estimate the bounding box and motion residuals.

## B.2 The Tracking Module

In this section, we provide the details of the tracking module that omit in the paper. Since DORT can estimate tightly coupled object location and motion, object tracking can be easily achieved via nearest center distances association [28, 29, 47]. Hence, our tracking module is mainly adapted from the previous distance-based object tracker [28, 29, 47]. Specifically, the tracking module contains four parts: Pre-processing, Association, Status Update and Life-cycle Management.

**Pre-processing** Given the detection results, the pre-processing stages mainly focus on filtering false negative objects. In our work, we first adopt Non-maximum Suppression to remove the duplicated bounding boxes with the threshold of 0.1 in terms of 3D IoU. Then we filter out the bounding boxes that the confidence threshold is lower than 0.25.

**Association** The association stage associates the tracked objects in frame t-1 and the detection results in frame t. We don't use the Kalman filter to predict the location of the trackers from frame t-1. Instead, we utilize the predicted velocity to propagate the detection results in frame t back to t-1. Then, we utilize the L2 distances of object centers to compute the similarity between the detected objects and the tracklets. Finally, the linear greedy matching strategy is adopted to achieve multi-object matching. **Status Update** After associating the detection results in frame t, we update the tracklets from frame t-1 into frame t. For the tracklets in frame t-1 that match with bounding boxes in frame t, we replace their object center location with the corresponding detection results to frame t. For the unmatched objects, we utilize the estimated object velocity to update its object center location to frame t. **Life-cycle Management** The life-cycle management module controls the "birth" and "depth" of the tracklets (*i.e.* birth, depth). Specifically, for the unmatched bounding boxes, they will be initialized as new tracklets. For the unmatched tracklets, we remove them when they are consecutive unmatched more than 2 times.

**Details of depth error calculation in Fig 2.** The depth error in Fig2 is calculated as follows. The depth error is the l1 distance between the ground truth depth and the depth obtained by assuming objects are static across frames. Specifically, the object depth that assumes objects are static is calculated as follows. We first utilize the ego-motion (camera extrinsic) and the ground-truth 2D location of 3D box centers in the past and current frames to obtain the cross-frame correspondence. Then, we utilize Eq2 to obtain the corresponding object depth.

## C More results on the Waymo dataset

In this section, we further provide the experimental results on the Waymo dataset for reference. We adopt ResNet-101 as the image backbone and train the model with 1/3 training data. Regarding initialization methods other than ImageNet pre-trained weights, we present additional results using the commonly used FCOS3D++ pre-trained weights on the Waymo dataset.

## D Ablation Studies

In this section, we provide the additional ablation studies that are omitted in the main paper. **DORT with Different Proposal Detector** We first show that DORT is agnostic with different proposal detectors (*e.g.* PGD [4], BEVDepth [23]). In Table 4, we display the experimental results of DORT with using PGD and BEVDepth as the proposal detectors. We can observe that the DORT is insensitive to the proposal detector and can consistently improve BEVDepth. We Benefiting from the low computation overhead of BEVDepth in the perspective part and the designed local volume, DORT also can achieve a more lightweight pipeline for dynamic object modeling.

Table 3: Experimental results on the Waymo validation set. ImageNet pre-trained denotes the supervised classification pre-training for the network backbone. FCOS3D++ on Waymo denotes the further pre-training of the model backbone via FCOS3D++ on the Waymo training dataset.

| Method | Pre-trained | mAPL | mAP |
|--------|-------------|------|-----|
| FCOS3D++ [15] | | 20.4 | 28.6 |
| DETR3D [24] | | 26.1 | 39.0 |
| BEVDepth [23] | ImageNet | 28.2 | 39.9 |
| MV-FCOS3D++ [51] | | 28.7 | 39.9 |
| Ours | | 30.1 | 42.3 |
| MV-FCOS3D++ [51] | FCOS3D++ | 33.8 | 46.7 |
| Ours | on Waymo | 35.0 | 48.9 |

Table 4: Experimental results on the nuScenes validation set. 1 past frame is adopted in the temporal modeling. $^*$ denotes the BEV FLOPS from the proposal detector.

| Method | mAP | NDS | Flops | |
|--------|-----|-----|-------|---|
| | | | PV | BEV |
| BEVDepth | 35.1 | 47.5 | 120.4 | 94.5 |
| DORT with PGD | 37.9 | 52.1 | 238.2 | 40.2 |
| DORT with BEVDepth | **38.1** | **52.1** | 120.4 | 74.4*+40.2 |

**Tracking with Semantic Embedding or Geometry Distance**  In this work, DORT achieves 3D object tracking via the nearest centerness association. To have a more comprehensive comparison of the tracking pipeline designed, we further provide the comparison of DORT by using semantic embedding to associate objects. Specifically, we follow previous methods [34] and adopt the widely-used quasi-dense similarity learning [52] to learn the tracking embedding. We extract two kinds of embedding features, one is from the perspective-view (PV) and another is from the bird-eye-view (BEV). In Table 5 and 6, we display the tracking results on the nuScenes tracking set. We can observe that DORT with geometry distance association can outperform the embedding-based methods by a large margin. Furthermore, it is also much simpler and more efficient and does not need to maintain an extra object embedding. Besides, the PV embedding is worse than the BEV-based embedding, which may be due to the view change in different cameras.

Table 5: Experimental results on the nuScenes validation set. 1 past frame is adopted in the temporal modeling.

| Method | AMOTA↑ | AMOTP↓ | MOTAR↑ |
|--------|--------|--------|--------|
| PV-Embedding | 36.8 | 1.412 | 44.2 |
| BEV-Embedding | 40.1 | 1.356 | 46.7 |
| DORT (Geometry Distance) | **42.4** | **1.264** | **49.2** |

# E  Theoretical Analysis of Ignoring Object Motion

In the main paper, we have shown that when ignoring object motion, the temporal correspondence would derive a biased depth. In this supplementary, we provide the full details of how ignoring object motion introduces a biased depth. We denote the camera intrinsic as $K$ and the ego-motion from frame $t_0$ to frame $t_1$ as $T_{t_0 \to t_1}^{ego}$:

$$K = \begin{bmatrix} f & 0 & c_u \\ 0 & f & c_v \\ 0 & 0 & 1 \end{bmatrix}, T_{t_0 \to t_1}^{ego} = \begin{bmatrix} 1 & 0 & 0 & x^{ego} \\ 0 & 1 & 0 & 0 \\ 0 & 0 & 1 & z^{ego} \end{bmatrix}. \tag{8}$$

Here, $f$ is the camera's focal length, and $(c_u, c_v)$ is the camera center coordinates in the image. For simplicity, we assume the ego-motion only contains the translation $(x^{ego}, 0, z^{ego})$ on the horizontal plane. The analysis also can be easily extended to a more complicated case that the motion contains rotation. Given the multiple-view images, temporal-based methods can utilize photometric or feature-metric similarity to find the correspondence of pixel $p_{t_0} = (u_{t_0}, v_{t_0})$ in the past frame $t_0$ and the pixel $p_{t_1} = (u_{t_1}, v_{t_1})$ in the current frame $t_1$.

Table 6: 3D object tracking results on the nuScenes validation set. We adopt ResNet-50 as the backbone and set the input resolution as $704 \times 256$.

| Method | AMOTA↑ | AMOTP↓ | Recall↑ |
|--------|--------|--------|---------|
| QD-Track3D [34] | 24.2 | 1.518 | 39.9 |
| Time3D [35] | 21.4 | 1.360 | N/A |
| TripletTrack [53] | 28.5 | 1.485 | N/A |
| MUTR3D [44] | 29.4 | 1.498 | 42.7 |
| QTrack [32] | 34.7 | 1.347 | 46.2 |
| DORT | **42.4** | **1.264** | 49.2 |

When we ignore the object motion, the depth $z_{t_1}$ of pixel $p_{t_1}$ can be recovered as:

$$T_{t_0 \to t_1}^{ego} \cdot \pi(p_{t_0}, K) = \pi(p_{t_1}, K),$$
$$z_{t_1} \frac{u_{t_1} + c_u}{f} - x^{ego} = \frac{u_{t_0} + c_u}{f}(z_{t_1} - z^{ego}),$$
$$z_{t_1} = \frac{z^{ego}(u_{t_0} - c_u) - fx^{ego}}{u_{t_0} - u_{t_1}}, \tag{9}$$

where $\pi$ denotes the projection from 2D image coordinate to 3D camera coordinate.

But as we showed in the main paper, the moving objects occupy large ratios in the driving scenarios. For example, when the object contains the translation $(x^{obj}, 0, z^{obj})$ in the horizontal plane, the object's motion can be represented as

$$T_{i \to j}^{obj} = \begin{bmatrix} 1 & 0 & 0 & x^{obj} \\ 0 & 1 & 0 & 0 \\ 0 & 0 & 1 & z^{obj} \end{bmatrix}. \tag{10}$$

With the object motion, the depth $z_{t_1}$ of pixel $p_{t_1}$ is recovered as:

$$T_{t_0 \to t_1}^{obj} T_{t_0 \to t_1}^{ego} \cdot \pi(p_{t_0}, K) = \pi(p_{t_1}, K),$$
$$z_{t_1} \frac{u_{t_1} + c_u}{f} - x^{ego} - x^{obj} = \frac{u_{t_0} + c_u}{f}(z_{t_1} - z^{ego} - z^{obj})$$
$$\hat{z}_{t_1} = \frac{(z^{ego} + z^{obj})(u_{t_0} - c_u) - f(x^{ego} + x^{obj})}{u_{t_0} - u_{t_1}}. \tag{11}$$

From Eq (9) and Eq (11), we can obtain the depth gap for the temporal correspondence with and without considering object motion:

$$\Delta z = \frac{z^{obj}(u_{t_0} - c_u) - fx^{ego}}{u_{t_0} - u_{t_1}}. \tag{12}$$

In Figure 6, we also provide a toy example to illustrate that one temporal correspondence can come from multiple combinations of object depth and motion (*i.e.* inaccurate depth with zero motion and accurate depth and GT motion). This means that if we inaccurately assume that objects are static across frames, the temporal correspondence would derive a misleading depth.

### E.1 Ill-posed Problem of Simultaneously Estimating 3D Location and Motion

Although object motion plays a critical role in temporal correspondence, however, it is non-trivial to estimate it from the monocular images. As shown in Figure 6, the one correspondence can come from infinite combinations of location and motion (the location can be the point in the ray $\overrightarrow{O_{t_0} P_{t_0}}$ and $\overrightarrow{O_{t_1} P_{t_1}}$, and the motion can be the line that connects the points.) Hence, it is an ill-posed problem that simultaneously estimates the 3D location and motion from the monocular images. To alleviate this issue, we leverage the rigid-body assumption for the objects in the driving scenarios and elaborate more temporal frames with constant velocity regularization to further constrain the motion.

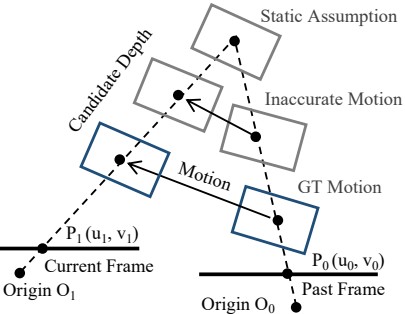

Figure 6: Different object motions can make the same temporal correspondence derive different depth.

# F  More Related Work

**Multi-View 3D Perception**  Leveraging multi-view images to recover 3D information is a fundamental topic, such as structure from motion [54], multi-view stereo [55], simultaneous localization and mapping [56], etc. One line of methods develop neural-network-based cost volumes [55, 57, 58, 45, 39, 59] to construct cross-frame visual cues for 3D perception. Another line of methods [60, 61, 62] constructs geometry constraints and leverages optimization techniques to obtain a tight-coupled 3D structure. However, most of the work assumes the scene and objects are static, making them fail to handle the moving objects in driving scenarios.

