# OpenReview forum: "DORT: Modeling Dynamic Objects in Recurrent for Multi-Camera 3D Object Detection and Tracking"
_robot-learning.org/CoRL/2023/Conference — CoRL 2023 Poster_

### Official Review · Reviewer_RSf9 · 2023-06-23

**Confidence:** 3
**Originality:** Very Good
**Technical Quality:** Very Good
**Clarity Of Presentation:** Very Good
**Impact:** 4

**Recommendation:**

Weak Accept: I recommend accepting the paper, but will not argue for my recommendation if the majority of other reviewers have a different opinion.

**Review:**

Strengths:
* \+ the paper provides a better understanding of the necessity of working locally by estimating objects motion and location instead of globally.
* \+ the authors promise an open-source code.
* \+ the idea is rather simple and leads to better performance.
* \+ computationally more efficient than global-features methods.
* \+ convincing ablation study (+appendix) on the NuScenes dataset.

Weaknesses/Limitations:
* \- limited evaluation setting, the model is only evaluated on a single dataset.
* \- no demonstration in a robotic domain.
* \- an additional limitation that is not addressed is handling occlusions.

Minor Comments:
* I think Figure 1 should be refined, as it is hard to understand from it the difference between previous methods and the proposed one. Also, in lines 132-134, the authors mention that this figure illustrates that “the correspondence of two points can come from infinite combinations of object location and motion” - I’m not confident this is indeed demonstrated in figure in its current form.
* Equation 4: I think $fx^{\text{ego}}$ should be $fx^{\text{obj}}$ in the resultant $\Delta z$.

Post Rebuttal:

I acknowldege that I have read the authors' rebuttal and the other reviewes. Thank you for the additional details and for addressing my concerns, I feel that most of them were answered. I'll keep my score, but as the other reviewers also raised important concerns, I will not argue my case if the other reviewers reach a different recommendation.



**Quality Of The Limitations Section:**

Additional details required

**Questions For Rebuttal:**

* In Figure 2, how is the empirical depth error calculated?
* In addition to the graphs in Figure 5, what is the performance-computation complexity trade-off w.r.t the number of refinement iterations $k$?
* While initializing with pre-trained ImageNet weights is a legitimate approach, have you experimented with training from scratch or using pre-trained self-supervised weights?
* Regarding the limitation I raised in “Weaknesses” above regarding occlusions, do you have an experiment/intuition how your model will handle occlusions? Are there other limitations inherited from previous methods?


**Robotics Focus:**

Relevant but unlikely to deploy to hardware in near future

**Summary Of Paper:**

The authors propose DORT, a method that estimates both object motion and location which are recurrently refined for better 3D object detection and tracking. They demonstrate empirically and theoretically that ignoring objects, especially when the motion varies between objects, leads to a biased depth estimation, harming the performance detectors.

**Summary Of Recommendation:**

I’m fairly convinced with the claims in this paper of the necessity of object motion estimation for better detection and tracking. While this seems applicable in robotics, the main application in this paper is for driving scenarios and the evaluation setting is limited to a single dataset. Overall the paper is clearly written and well-motivated and I believe that extending the evaluation setting will benefit this paper, in addition to answering my questions under “Questions for Rebuttal” and additional discussion on limitations (even if it ends up in the Appendix).

Post Rebuttal: I feel that most of them were answered.

---

### Official Review · Reviewer_VMsm · 2023-07-19

**Confidence:** 4
**Originality:** Good
**Technical Quality:** Very Good
**Clarity Of Presentation:** Very Good
**Impact:** 3

**Recommendation:**

Weak Accept: I recommend accepting the paper, but will not argue for my recommendation if the majority of other reviewers have a different opinion.

**Review:**

Strength

* The paper is well written and easily conveys its contributions. The descriptions of the object-wise local volume and recurrent dynamic object modeling are clear and intuitive.The algorithm elegantly combines two-stage detectors with recurrent dynamic modeling. Also, the tracking variant is quite seamlessly built from the detectors. This algorithm design leads me to wonder how it performs in an online setting? Also, can it be easily changed into an autoregressive way?

* Some experimental designs are quite interesting. For example, the experiment shown in Figure-5 validates both the effectiveness of the feature aggregation strategy and the motion prediction.

* The experimental results appear strong. While browsing over the leaderboard of nuScenes detection and tracking, it is evident that the current algorithm achieves SOTA results under the ConvNeXt-B or ResNet setting.

Weakness

* Although the experimental results look good, I believe the author might need to add more experiments to solidify the findings. This could include extending to larger backbones or adding the Waymo dataset. Larger backbones might be essential as the current leaderboard results are all based on them. Tests with large backbones can validate the scalability (including computational cost, memory cost, and performance growth) of the proposed algorithm.

* The paper does not list any metrics on parameters, computational cost, or memory cost, yet it claims efficiency using object-centric features rather than constructing an entire BEV. The author needs numerical evidence to support this claim.

* Another minor concern is about the first stage detector used in this paper. The author seems to use BEVDepth only; can the author provide more analysis on the first-stage detectors and their impact on the final performance?

**Quality Of The Limitations Section:**

Limitations are addressed clearly

**Questions For Rebuttal:**

Most of my questions are mentioned in the weakness section, but let me summarize them here:

* What is the performance of the proposed methods with larger backbones or on the Waymo dataset?
* Can the author provide more metrics about parameter count, memory cost, and computational cost (FLOPS might be sufficient if the author cannot measure the inference speed)?
* Can the author provide more analysis on how the first-stage detector affects the final results?
* I am also curious about whether the proposed method can be easily extended to an online setting. (It's okay if this question isn't answered; I'm just curious.)

**Robotics Focus:**

Relevant but unlikely to deploy to hardware in near future

**Summary Of Paper:**

This paper presents a 3D object detection and tracking algorithm with multiview videos in the scenarios of autonomous driving. The key improvement is the algorithm's ability to aggregate multiview features for each candidate object while accounting for object motions. The author compares the proposed method with multiple current SOTA methods like BEVDepth, Pseudo-LiDAR R-V2, and UV-RCNN on the nuScenes dataset, demonstrating impressive performance improvements of 1.1 NDS and 5.7 AMOTA.

**Summary Of Recommendation:**

This paper presents an interesting algorithm for multiview, multi-frame 3D object detection and tracking. It improves upon the current state-of-the-art (SOTA) on the nuScenes dataset. The paper is also well-written. Though I think there are a couple of improvements that can be made to this paper (as mentioned in the question part), but overall, I feel it is acceptable.

---

### Official Review · Reviewer_wd6g · 2023-07-19

**Confidence:** 3
**Originality:** Good
**Technical Quality:** Good
**Clarity Of Presentation:** Good
**Impact:** 3

**Recommendation:**

Weak Reject: I recommend rejecting the paper, but will not argue for my recommendation if the majority of other reviewers have a different opinion.

**Review:**

Overall, the paper is well structured. Some typos here and there could be improved. There are parts that are a bit unclear and this could be improved as well:
* Multiple Estimation Fusion: looking at Fig.3 it looks like fusion across time happens on a feature level. But then it is stated that fusion happens on a residual level. Could you clarify maybe?
* Further details on how tracking is done would be helpful (4.4). There is mention of a Kalman filter but details on prediction and update are missing. Also, in the tracking case, would one be able to get rid of the initial candidates and directly use the bounding boxes predicted by the filter?

I am not sure how much value the analysis in chapter 3 adds. It's fairly clear that when the static assumption is violated that there will be an estimation error. And that this estimation error is linear in speed and time is quite intuitive. I would recommend leaving this away and rather improving formatting (it looks very squeezed) and provide more details w.r.t. method and results.

Do you think the constant velocity assumption could be dropped and rather the location residual for all timesteps could be predicted? This would still require proper alignment of the bounding boxes across time frames. The geometric constraint across time would be loosened up, but features would still be shared across time and this would avoid errors caused when the velocity is not constant?

The employed method seems adequate and the results are competitive. My only concern is that there is no big novelty. Introducing velocity estimation for moving targets has been done before in 3D vision, e.g. even with moving per-object volumes (Xu 2019 - Mid-Fusion). There is still value in putting everything together though. Another small shortcoming is the limitation to planar motion and I am not even sure that a rotational component is included in the velocity. If, as the introduction claims, this is interesting for other domains than autonomous driving, then it would be interesting in knowing what happens if the velocity constraints are relaxed and estimation is done on the full 6 DOF.

**Quality Of The Limitations Section:**

Limitations are addressed clearly

**Questions For Rebuttal:**

* Clarify above questions
* Remove/shorten chapter 3 and relax formatting
* Discuss what happens if velocity constraints are relaxed

**Robotics Focus:**

Highly relevant to robotics but no hardware experiments

**Summary Of Paper:**

The paper introduces a multi-view object detection and tracking framework that is able to deal with dynamic objects. To this end it introduces per-object cost volumes and parameterizes their location as a function of time. A constant velocity model is assumed and the corresponding location and velocity parameters are refined iteratively. The multi-view object tracking results are competitive.

**Summary Of Recommendation:**

The paper is technically sound but the contribution is rather incremental.

---

### Official Review · Reviewer_fWSU · 2023-07-23

**Confidence:** 3
**Originality:** Good
**Technical Quality:** Good
**Clarity Of Presentation:** Good
**Impact:** 4

**Recommendation:**

Weak Accept: I recommend accepting the paper, but will not argue for my recommendation if the majority of other reviewers have a different opinion.

**Review:**

The problem statement is well articulated with a good explanation in Section 3 with a nice visualization from Figure 2. The proposed method is technically sound and has proven to show better results in some of the metrics of detection and in all the metrics of tracking compared to SOTA.

Please find my comments for the authors to address in the rebuttal.

1. The proposed method is trained and tested on only the nuScenes dataset. I would like to see more results on the other dataset in the main paper.

2. The authors argue that the proposed method alleviates the heavy computational burden that comes with global BEV methods [lines 8 - 9 ]. I would like to see some experiments or discussions regarding this.

3. Also the authors suggest that the proposed method can be plugged into various existing methods [lines 12-13]. I am keen to see some experiments or discussion regarding that.

4. The authors attribute a 0.5% improvement in mAP as significant. Also, I am not able to get this relative improvement number (both 0.5% and 7.4% from line 275) from Table 1. Please explain the calculations.

5. What do the authors mean by "right-body movement" in Line 41, 135? Is it supposed to be rigid-body?

6. I think Figure 5 is mentioned as Table 5 in Line 294.

**Quality Of The Limitations Section:**

Limitations are addressed clearly

**Questions For Rebuttal:**

Please address the above-mentioned comments.

**Robotics Focus:**

Highly relevant to robotics but no hardware experiments

**Summary Of Paper:**

This paper proposes a multi-camera 3d object detection and tracking method. The SOTA methods use global feature volumes for this task with an added assumption of the scene of interest being static. This assumption fails for the scenes with dynamic objects. The authors propose to address this issue with object-wise local feature volumes and further estimate the motion and location of the objects using a recurrent module. The experiments were conducted on Nuscenes detection and tracking benchmarks where they were able to achieve 57.6% and 62.8% in AMOTA, and NDS metrics respectively.

**Summary Of Recommendation:**

This paper proposes to address the problem of dynamic objects in 3D objection detection and tracking using an approach that uses local-feature volumes and iterative refinement of object motion and bounding boxes using a recurrent module. The improvements of the detection module seem marginal but the tracking results are good in all the metrics. However, the current version of the paper needs some more experiments to validate their claims.


Post rebuttal, my concerns are addressed and I am happy to change the recommendation to weak accept.

---

### Author Response · Authors · 2023-08-11
**General response**

We would like to thank the reviewers for their thoughtful feedback. During rebuttal, we further provide additional experiments on the Waymo dataset, computational analysis of our local volume compared to the global volume, and a detailed analysis of our method plugged with different detectors. Please refer to the individual response for more details. Regarding the typo and writing suggestions, we will carefully consider them in the revision.

---

### Decision · Program_Chairs · 2023-08-30

**Decision:**

Accept (Poster)

**Comment:**

The paper proposes a a multi-camera 3D object detection and tracking method which leverages local volumes for motion estimation while accounting for object motions. The proposed method introduces some novelty in multi-view dynamic object tracking. The reviewers have mainly raised questions about the limited experimentation, handling of occlusions, list of metrics, etc. Most concerns have been addressed by the authors. The camera ready method should definitely include the additional experiments provided in the rebuttal with sufficient explanations, a description of experimental protocol and metrics, as well as, a broader limitations discussion based on the comments of the reviewers.